# The Vitamin D Receptor as a Prognostic Marker in Breast Cancer—A Cohort Study

**DOI:** 10.3390/nu16070931

**Published:** 2024-03-23

**Authors:** Linnea Huss, Igis Gulz-Haake, Emma Nilsson, Helga Tryggvadottir, Linn Nilsson, Björn Nodin, Karin Jirström, Karolin Isaksson, Helena Jernström

**Affiliations:** 1Department of Clinical Sciences, Division of Oncology, Lund University, 221 85 Lund, Sweden; igisgh@hotmail.com (I.G.-H.); emma.nilsson.4335@med.lu.se (E.N.); helga.tryggvadottir@med.lu.se (H.T.); linn.nilsson@med.lu.se (L.N.); bjorn.nodin@med.lu.se (B.N.); karin.jirstrom@med.lu.se (K.J.); 2Department of Surgery, Helsingborg Hospital, 251 87 Helsingborg, Sweden; 3Department of Hematology, Oncology, and Radiation Physics, Skåne University Hospital, 221 85 Lund, Sweden; 4Department of Medical Physics and Engineering, Växjö Central Hospital and Department of Research and Development, Region Kronoberg, 352 34 Växjö, Sweden; 5Department of Clinical Sciences, Division of Surgery, Lund University, 221 85 Lund, Sweden; karolin.isaksson@med.lu.se; 6Department of Surgery, Kristianstad Hospital, 291 85 Kristianstad, Sweden

**Keywords:** vitamin D receptor, breast cancer, prognosis, breast cancer free interval, overall survival

## Abstract

Previous research has indicated an association between the presence of the vitamin D receptor (VDR) in breast cancer tissue and a favorable prognosis. This study aimed to further evaluate the prognostic potential of VDR located in the nuclear membrane or nucleus (liganded). The VDR protein levels were analyzed using immunohistochemistry in tumor samples from 878 breast cancer patients from Lund, Sweden, included in the Breast Cancer and Blood Study (BCBlood) from October 2002 to June 2012. The follow-up for breast cancer events and overall survival was recorded until 30 June 2019. Univariable and multivariable survival analyses were conducted, both with complete case data and with missing data imputed using multiple imputation by chained equations (MICE). Tumor-specific positive nuclear membrane VDR^(num)^ staining was associated with favorable tumor characteristics and a longer breast cancer free interval (BCFI; HR: 0.64; 95% CI: 0.44–0.95) and overall survival (OS; HR: 0.52; 95% CI: 0.34–0.78). Further analyses indicated that VDR^num^ status also was predictive of overall survival when investigated in relation to ER status. There were significant interactions between VDR and invasive tumor size (P_interaction_ = 0.047), as well as mode of detection (P_interaction_ = 0.049). VDR^num^ was associated with a longer BCFI in patients with larger tumors (HR: 0.36; 95% CI: 0.14–0.93) or clinically detected tumors (HR: 0.28; 95% CI: 0.09–0.83), while no association was found for smaller tumors and screening-detected tumors. Further studies are suggested to confirm our results and to evaluate whether VDR should and could be used as a prognostic and targetable marker in breast cancer diagnostics.

## 1. Introduction

Breast cancer patients are more likely to have low serum levels of vitamin D than healthy women [1]. There are also studies indicating an association between a worse breast cancer prognosis and lower vitamin D levels [2,3] and that vitamin D supplementation reduces the risk of advanced cancer [4]. The active form of vitamin D, 1,25(OH)_2_D_3_, binds to the vitamin D receptor (VDR), which translocates to the nucleus and affects regulatory genes involved in cell growth, apoptosis, and cell signaling [5]. Several studies have investigated the associations between VDR in breast cancer tissue and breast cancer prognosis and found VDR to be associated with favorable outcomes [6,7,8,9,10,11,12]. These findings suggest that VDR could be used as a biomarker for tumor progression [13]. There is also one study based on in vitro and animal models indicating that unliganded VDR may stimulate tumor growth [14]. The subcellular location of VDR could thus be of importance, an observation that is not well captured using gene expression analyses.

The current study was initiated to further elucidate the prognostic potential of VDR in breast cancer. The primary aim was to validate previous research that has suggested a positive association between the presence of nuclear VDR and breast cancer prognosis. The secondary aim was to explore whether the subcellular localization of VDR yields refined prognostic information in breast cancer. Further, this study aimed to explore whether the associations between VDR and prognosis differ between subgroups of breast cancer.

## 2. Material and Methods

### 2.1. Study Population

The Breast Cancer and Blood Cohort study (BCBlood) was initiated in October 2002. Tumor samples from patients diagnosed with primary breast cancer up to June 2012 were assembled into a tissue micro array (TMA). Patients with a previous breast cancer or other cancer diagnosis within the last 10 years were excluded. During this period, 1116 patients were included in the study. Of these, 39 patients diagnosed with non-invasive disease, 51 patients who were recommended preoperative treatment, and 8 patients with an early breast cancer event within 0.3 years from diagnosis were excluded from the current study, leaving 1018 eligible patients. No tissue was available for TMA construction in 34 cases, and tumor samples from another 106 cases were non-evaluable for VDR. A flowchart of the included and excluded patients is presented in Figure 1.

### 2.2. Clinical Information and Histopathological Analysis

Patients were included at their preoperative visit and completed a questionnaire regarding medication (including vitamin supplements) taken during the past week, lifestyle, and reproductive factors [15]. The vitamin D content in the supplements was obtained from the products’ home pages. Supplements reported as multivitamins were considered to include vitamin D since most Swedish brands of multivitamins contain vitamin D. Any unspecified “calcium-supplement” was not considered to include vitamin D. 

Anthropometric measurements including height, weight, and waist and hip circumference were taken by research nurses. A body mass index (BMI) of 25 kg/m^2^ or higher was considered overweight according to the definition of overweight from the World Health Organization (WHO) [16]. A waist circumference equal to or above 80 cm in females was used as a marker for central overweight/obesity, in agreement with WHO [17]. Breast volume was calculated as the sum of the right and left breasts, measured using differently sized plastic cups [18]. A breast volume equal to or above 850 mL was used as the cut-off, as suggested by a previous study [19]. Clinical information, including adjuvant treatment information, was collected from both questionnaires and medical records. 

Tumor characteristics, such as tumor size, axillary lymph node involvement, histologic grade and type, and estrogen receptor (ER) and progesterone receptor (PR) status, were collected from pathology reports. Human epidermal growth factor receptor 2 (HER2) amplification was introduced into clinical practice in November 2005. A retrospective HER2 analysis using dual gene protein staining was performed on the TMA from patients included in 2002–2012 [20], and pathological assessment of HER2 status was complemented with the data from the TMA. There was a 97.7% agreement between the two methods.

### 2.3. Tissue Microarray

The TMA was constructed at the Department of Pathology in Lund in 2013. Duplicate 1 mm cores from formalin-fixed paraffin-embedded tissue blocks of representative tumor tissue were collected and assembled into a recipient TMA block using a semi-automated tissue array device (Beecher Instruments Inc., Sun Prairie, WI, USA). Sections of 4 μm were placed on glass slides and stored at −20 °C until immunohistochemical staining. Deparaffinization and antigen retrieval were performed using an automatic PT Link system (Agilent Technologies, Santa Clara, CA, USA). Immunohistochemistry was performed using the Autostainer Plus with the EnVision FLEX high-pH kit, according to the manufacturer’s instructions (Agilent Technologies). A previously well-validated mouse monoclonal D-6 antibody (sc-13133 Santa Cruz Biotechnology, CA, USA; diluted 1:750) was selected for the VDR staining due to its high sensitivity and specificity [21]. The dilution was optimized on a selection of tissue samples from different breast cancer subtypes to obtain a good separation between clearly negative results (blue) and strong intensity (brown) (Images in Figure 1 and Figure 2).

### 2.4. Microscopy Assessment

The two cores of each tumor were microscopically assessed jointly, first simultaneously by two junior investigators (I.G.H., E.N.) and again by a more experienced investigator (L.H.). The observers were blinded to any patient data but had access to information on whether the core represented an invasive tumor.

VDR staining was assessed in invasive tumor cells in different subcellular locations, in the cytoplasm (VDR^cyt^), in the nuclear membrane (VDR^num^), and in the nucleus (VDR^nuc^). The cytoplasmic intensity was graded as negative (0), weak (1+), moderate (2+), or strong (3+) if at least 20% of the tumor cells had the highest graded intensity. To facilitate the calculations, tumors with a negative or weak cytoplasmic intensity were merged into VDR^cyt^low (*n* = 270) and moderate or strong cytoplasmic intensity into VDR^cyt^high (*n* = 608). A nuclear membrane was scored as positive if at least 10% of the cells had stained nuclear membranes. Nuclear staining of 10% or more was not observed.

A total of 24 patients had bilateral tumors, of whom 19 had evaluable tumors from both sides. The tumor with the highest cytoplasmic score for VDR was used in case of discrepancy, and all the other tumor characteristics were taken from the corresponding side. Since only four tumors were discordant for VDR^num^, no sensitivity analysis was carried out.

Out of the tumors included in the TMA, 106 were not evaluable for VDR for various reasons. For example, the cores may have been dislocated, or there were too few or no invasive tumor cells in the cores.

### 2.5. Endpoint Retrieval

The end of follow-up was 30 June 2019. Information on survival was retrieved from the Swedish Population Registry, and information on breast cancer events was obtained from the Regional Tumor Registry, pathology reports, and medical records. Patients who emigrated were followed until the last follow-up before emigration. The endpoints in the breast cancer free interval (BCFI) analyses were local or regional recurrence, contralateral breast cancer, or distant metastasis, and the follow-up time was censored at the last follow-up or death due to any cause. Death of any cause was the endpoint in the overall survival (OS) analyses.

### 2.6. Statistical Methods

SPSS^®^ version 28 (IBM Corp., Armonk, NY, USA) was used for the statistical analyses. Due to the results of a previous study [8,9], the statistical analyses primarily focused on associations with VDR^num^. The patient and tumor characteristics as well as the treatment factors were transformed into categorical variables and compared using a Chi^2^ test between the categories of VDR^num^. 

The correlations between the staining intensity and the time between surgery and staining (years) were calculated using Spearman’s Rho (R_s_). The staining intensity of VDR^num^ was correlated with the time between surgery and staining (*p* < 0.001), and the logistic regression models and other statistical analyses were therefore adjusted accordingly. Kaplan–Meier estimates were plotted to visualize univariable associations between VDR^num^ and BCFI or OS. 

Cox proportional hazards models yielding hazard ratios (HRs) and 95% confidence intervals (CIs) were constructed in order to evaluate the associations between VDR^num^ and BCFI and OS, taking confounding factors into account. Model 1 was adjusted only for the time between surgery and staining. Model 2 included Model 1 and was further adjusted for age (continuous), invasive tumor size (pT2/3/4), axillary lymph node status (pN+), and grade III. Model 3 was adjusted for the same factors as Model 2, with further adjustment for ER status. Model 4 included Models 1–3 and was further adjusted for a BMI ≥ 25 kg/m^2^, smoking, and adjuvant treatments (chemotherapy, radiotherapy, tamoxifen, and aromatase inhibitors (AIs)).

The multivariable analyses were first performed using cases with complete data for all variables (complete case analysis) and thereafter including all 1018 patients included in the study, after multiple imputation of missing data. The imputation model is described in detail in Appendix B.

To evaluate whether VDR could be used as a prognostic factor in patients with ER-positive breast cancer, the tumors were divided into three groups: (1) ERposVDR^num^pos (*n* = 214), (2) ERposVDR^num^neg (*n* = 555), (3) ERnegVDR^num^neg (*n* = 107). There was only one ER-negative tumor with positive VDR^num^, and this tumor was excluded from this analysis.

To examine the effect modifications of the prognostic impact of VDR^num^ according to clinicopathological factors (age at inclusion ≥50 years, preoperative BMI ≥ 25 kg/m^2^, waist circumference ≥80 cm, current smoking, alcohol abstention, vitamin D supplementation, mode of detection, invasive tumor size > 20 mm or skin or muscular involvement, positive lymph node status, Nottingham grade III, ER status, chemotherapy, radiotherapy, tamoxifen, and AIs), multiplicative interaction variables between these investigated factors and VDR^num^ were calculated. The interaction analyses were adjusted according to Model 3. 

All the *p*-values were 2-tailed, and a *p*-value < 0.05 was considered statistically significant. Nominal *p*-values without adjustment for multiple testing are presented.

## 3. Results

Out of the 984 patients with tumors in the TMA, VDR was evaluable for 878 tumors. The tumors not included in the TMA were generally smaller and more often lymph-node-negative and of a lower grade than the tumors included in the TMA (Table 1).

When evaluated for cytoplasmic intensity, 62 (7%) were negative for VDR, 208 (24%) were weak, 496 (56%) were moderate, and 112 (13%) were strong (Appendix A). VDR^num^ positivity was present in 215 (25%) patients. 

Positive nuclear membranous VDR (VDR^num^pos) was seen almost exclusively in tumors with a high cytoplasmic intensity, but 398 tumors with a high cytoplasmic intensity were evaluated as VDR-negative in the nuclear membrane (VDR^num^neg).

Four groups of VDR distribution in different subcellular locations were created: (1) VDR^cyt^lowVDR^num^pos (*n* = 5), (2) VDR^cyt^highVDR^num^pos (*n* = 210), (3) VDR^cyt^highVDR^num^neg (*n* = 398), and (4) VDR^cyt^lowVDR^num^neg (*n* = 265) (Figure 2).

### 3.1. Patient Characteristics

Table 2 presents a distribution of the patient characteristics in relation to VDR^num^ status. The median age at inclusion was 61 years with a range of 24–99 years. Patients with VDR^num^pos tumors had statistically significantly smaller waist circumferences (*p* = 0.041) and breast volumes than patients with VDR^num^neg tumors (*p* = 0.050), and their tumors were screening-detected to a larger extent (*p* = 0.033). No other associations with patient characteristics reached statistical significance. A slightly larger proportion of VDR^num^neg tumors was operated on during fall, but this association was not statistically significant (*p >* 0.3).

### 3.2. Tumor Characteristics and Adjuvant Treatment

VDR^num^pos tumors were associated with a lower grade, with positive hormone receptors (ER and PR), and with tumors not amplified for HER2 (all *P_s_* < 0.001). The tumors evaluated as VDR^num^pos were also more often smaller than 20 mm (*p* = 0.060), although this association was not statistically significant (Table 1).

After surgery, a larger proportion of patients with VDR^num^neg tumors received adjuvant chemotherapy (*p <* 0.001) and trastuzumab (*p* = 0.035). No difference was seen regarding radiotherapy between patients with VDR^num^pos and VDR^num^neg tumors. The patients with VDR^num^neg tumors more often received adjuvant endocrine therapy, both as tamoxifen (*p* < 0.001) and AIs (*p* = 0.060), compared with the patients with VDR^num^pos tumors (Table 1).

Appendix A presents the distribution of the patient characteristics, tumor characteristics, treatment factors, and events in relation to cytoplasmic VDR levels. 

### 3.3. VDR Levels and Prognosis

Patients were followed with questionnaires up to 15 years after inclusion. At the end of the follow-up, 735 of the 1018 patients were still at risk, and the median follow-up time was 9.0 years (IQR 7.0–11.1) for these patients. During follow-up, 195 of the patients experienced a breast cancer event, and 188 patients died, of whom 100 patients had a previous breast cancer event.

Patients with VDR^num^-positive tumors had a better prognosis, both in terms of BCFI and OS, in the univariable analysis (Figure 3A,B), and these associations remained significant in the multivariable analyses for both outcomes (Table 3). Statistical significance was seen in both the complete case analyses and pooled analyses after multiple imputation of the BCFI (HR 0.65 (95% CI 0.43–0.99)) and OS (HR 0.54 (0.35–0.85)), only adjusted for the time between surgery and staining. After further adjustment for other known prognostic factors (age, tumor size, node status, grade III, ER status), the results were no longer statistically significant for the BCFI (HR 0.75 (0.50–1.12)), whereas the association of VDR^num^pos tumors with a longer OS remained significant: HR 0.64 (0.41–0.99).

When patients were categorized into four groups depending on the subcellular localization of VDR, the univariable analyses showed the best prognosis for the VDR^num^pos tumors, irrespective of VDR^cyt^ (Figure 3C,D). There were no events and there was only one death in the group with VDR^cyt^lowVDR^num^pos, but this group was very small (*n* = 5). 

The univariable analyses of combined ER and VDR^num^ status showed that patients with ERposVDR^num^pos tumors had the best prognosis in terms of both BCFI and OS (Figure 3E,F). The least favorable prognosis was seen for patients with ERnegVDR^num^neg tumors. Appendix A presents the distribution of the clinicopathological factors in relation to VDR^num^ only for the ERpos tumors and demonstrates that the VDR^num^pos tumors, in this context, still were associated with a lower frequency of Nottingham grade III tumors (*p* < 0.001). The Cox regression analyses also showed the longest OS for patients with ERposVDR^num^pos tumors in models adjusted for prognostic factors (Appendix A). 

### 3.4. Effect Modification between VDR^num^ and Clinicopathological Factors

The formal interaction analyses (conducted without the use of multiple imputation of missing data) showed an interaction between VDR^num^ and invasive tumor size for BCFI (adjusted HR model 3 = 0.35; *p* = 0.047), as well as between VDR^num^ and the mode of detection (analysis restricted to patients in screening age 45–74 years) in relation to BCFI (adjusted HR interaction model 3 = 3.58, *p* = 0.049). 

Further stratified analyses showed that patients with larger tumors (>20 mm or with skin or muscular involvement) that were VDR^num^pos had significantly fewer breast cancer events (adjusted HR model 1 = 0.36 (0.14–0.93), an association that was not observed for patients with smaller tumors (HR 0.85 (0.53–1.35) (Figure 4A,B). Also, patients with clinically detected breast cancer had a better prognosis with regards to BCFI if the tumors were VDR^num^pos (HR 0.28 (0.09–0.83)), an association that was not seen for the screening-detected tumors (HR 0.94 (0.57–1.54)) (Figure 4C,D). 

No effect modifications were found in the interaction analyses between age at diagnosis, BMI, waist circumference, smoking, alcohol intake, vitamin D supplementation, lymph node status, Nottingham grade, chemotherapy, radiotherapy, tamoxifen, aromatase inhibitors, and VDR^num^ in relation to BCFI, and no interaction at all could be shown for any of the tested clinicopathological factors in relation to OS.

The interaction between VDR^num^ and ER status could not be tested since there was only one VDR^num^pos tumor that was evaluated as ER-negative.

## 4. Discussion

The main findings of this study were significant associations between positive VDR staining of the nuclear membrane in breast cancer with favorable tumor characteristics and a longer BCFI and OS. The results also showed that evaluation of the nuclear membrane levels of VDR was a better predictor of prognosis than cytoplasmic levels. It is likely that nuclear membrane VDR represents liganded VDR, whereas VDR in the cytoplasm represents unliganded VDR [14]. Furthermore, the results suggest that VDR^num^ status may be used to further refine the selection of luminal breast cancers with a favorable prognosis. This study also found that the prognostic value of VDR^num^ was impacted by the interactions between VDR^num^ status and tumor size, as well as the mode of detection. 

### 4.1. Associations of VDR with Different Types of Breast Cancer

This study further confirms the results of several previous studies, demonstrating that VDR expression in breast cancer tissue is inversely associated with disease aggressiveness [8,22,23]. The low levels of VDR in TNBC and HER2-amplified breast cancer could in part be explained by the VDR gene being a target of the tumor suppressor gene p53 and its family members [24]. *TP53* mutations are most common in TNBC (80%) and HER2-amplified cancers (70%), and their frequency is much lower in Luminal A (10%) and Luminal B type cancers (30%) [25].

Another study has shown that positive VDR staining in TNBC is correlated inversely with a high mitotic score, histologic grade, and Ki67 and directly with an increased OS and suggested that VDR could be a possible therapeutic target in TNBC [26]. In our study, 62 tumors were evaluated as VDR^cyt^-negative, and more than half (56.6%) of these tumors were classified as triple-negative, which meant that 49.4% out of the 74 TNBC cases in the cohort were also VDR-negative. As noted above, this implies that such tumors may also have a mutated p53 gene [25]. In a study on TNBC cell lines, it was shown that mutated p53 interacted with VDR so that local vitamin D (1α25(OH)_2_D_3_), instead of functioning as a pro-apoptotic agent, was converted into an anti-apoptotic agent [27]. Taking this into consideration, further research on the interactions between the VDR and vitamin D levels in the TNBC subgroup should be of particular interest.

It might be assumed that all the associations between VDR and a favorable prognosis could be due to a correlation between VDR and ER, but the subgroup analysis restricted to ER-positive breast cancer indicates that this is not the case. The patients with ERpos tumors who were also VDR^num^-positive had a better prognosis, with fewer breast cancer events at a long follow-up and a longer OS. Furthermore, other studies have shown similar results, with nuclear VDR being associated with decreased breast cancer mortality for patients with Luminal B-like tumors [8] and a positive association between VDR and a longer disease-free survival for patients with Luminal A tumors [11]. Therefore, VDR^num^ status may be used to refine the prognostics of luminal breast cancers and contribute to the identification of patients recommended to receive adjuvant treatments.

According to the interaction analyses in this study, we found VDR^num^ status to be predictive mainly of larger tumors (pT2/3/4) and clinically detected tumors. Naturally, larger tumors are more often clinically detected, and therefore similar results in the interaction analyses strengthen the validity of these results. Also, we found an interaction between a single-nucleotide polymorphism (SNP) of Fok1 (located in the *VDR* gene) and tumor size in another study conducted by our study group [28]. Although there was no statistically significant association between Fok1 and VDR^num^ status, the results strengthen the notion that the VDR signaling pathway interacts with tumor progression.

### 4.2. Using VDR as a Prognostic Factor and Vitamin D Supplements in Breast Cancer Treatment 

This study used a previously well-validated antibody [21,29], and the TMA was constructed and stained by an experienced biomedical analysist (B.N.) after a senior pathologist (K.J.) indicated from which areas the cores should be obtained. All the cores were assessed twice by separate readers, and in the case of a discrepancy between the readers, the senior evaluator re-evaluated the tumor cores and decided on a final score. Due to the lack of a validated standardized evaluation method for VDR, direct comparisons with the results from similar previous studies are difficult.

Hence, before VDR can be introduced as a prognostic marker, there are still several questions to be answered. First of all, when it comes to the potential clinical relevance of such an analysis, our results suggest that positive VDR^num^ is associated with a favorable prognosis also when tumors are compared within molecular subgroups, but more studies are needed to confirm these results. Secondly, a standardized method for analysis in the clinical setting needs to be developed. One could also ask whether it would be of greater importance to evaluate the VDR expression in the nuclear compartments, where it appears to carry a prognostic value, or in the cytoplasm, where it could be an indicator of the therapeutic effect of vitamin D supplementation.

Since breast cancer diagnostics today not only relies on microscopic evaluation by pathologists, other methods besides immunohistochemistry, such as gene expression analysis, should be further explored for associations between the VDR and breast cancer outcomes.

The subcellular distribution of VDR could be influenced by its saturation with 1,25(OH)_2_D_3_ [14]. In this study, we could not find any associations between either vitamin D supplements being taken at the time of breast cancer diagnosis or season of operation and VDR^num^, although both are known to influence serum levels of vitamin D. Another previous study examining the pre-diagnostic serum levels of 25OHD in association with the nuclear VDR expression in subsequent breast cancer could not find any statistically significant associations, although it reported a tendency towards a lower risk of VDR-negative breast cancer with high vitamin D levels [9]. It might be of importance to investigate whether there is such an association since it is probable that a shifting subcellular location of VDR towards the nucleus is favorable for breast cancer prognosis. Therefore, further research is also needed to find associations between the factors that potentially influence vitamin D levels and the subcellular location of VDR. 

### 4.3. Generalizability of the Results

Patients with previous breast cancer or another cancer diagnosis within the last 10 years were not included in the BCBlood cohort. Due to a lack of research nurses, not all the breast cancer patients at Skåne University Hospital in Lund could be included in the cohort. Therefore, the multiple imputation analysis does not include the whole population of breast cancer patients. Despite this, BCBlood is considered representative of breast cancer patients who have been treated in Lund [15]. Most of the patients included in the cohort were of Swedish or European decent (although no data on ethnicity were collected), and it is therefore uncertain whether the findings can be applied to a heterogenous population of breast cancer patients. 

This study was conducted and presented in congruence with the guidelines REMARK and STROBE for cohort studies [30,31]. 

## 5. Conclusions

In conclusion, positive VDR staining in the nuclear membranes of invasive breast cancer cells was associated with an improved breast cancer prognosis. VDR could potentially be used to refine the prognostication for ER-positive breast cancer. The prognostic potential of VDR seemed to be confined to large and/or clinically detected breast cancers. There are still several questions to be answered about the interplay between vitamin D, the vitamin D receptor, and how it can be of use in breast cancer prognostication and treatment.

## Figures and Tables

**Figure 1 nutrients-16-00931-f001:**
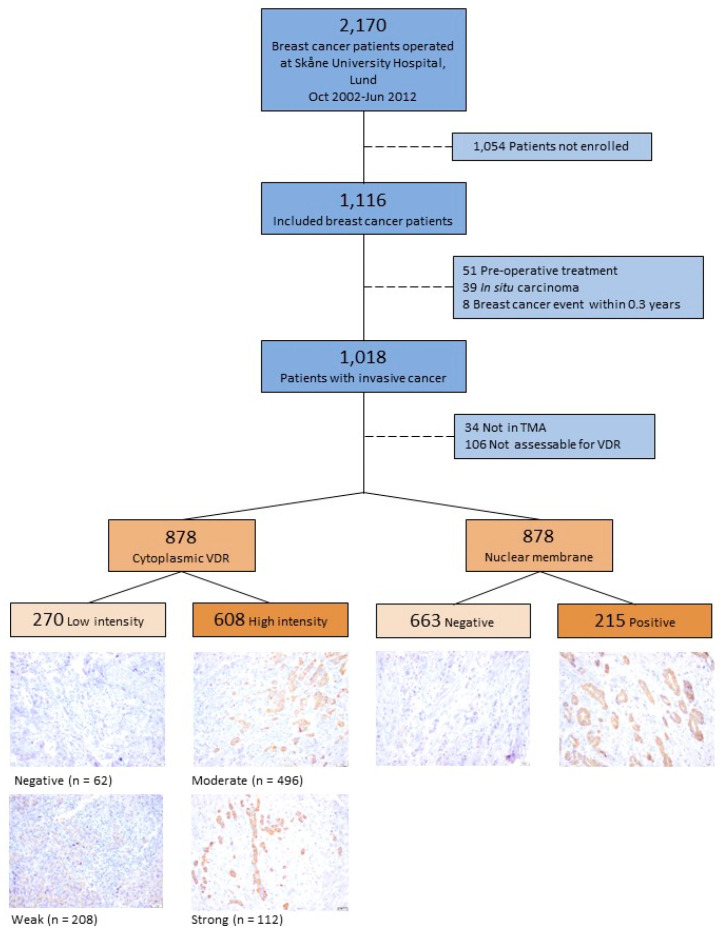
Flowchart of inclusion and exclusion of patients and representative images of immunohistochemical staining intensities of nuclear membrane and cytoplasmic VDR (40×) in breast cancer tissue. The bar represents 20 μm.

**Figure 2 nutrients-16-00931-f002:**
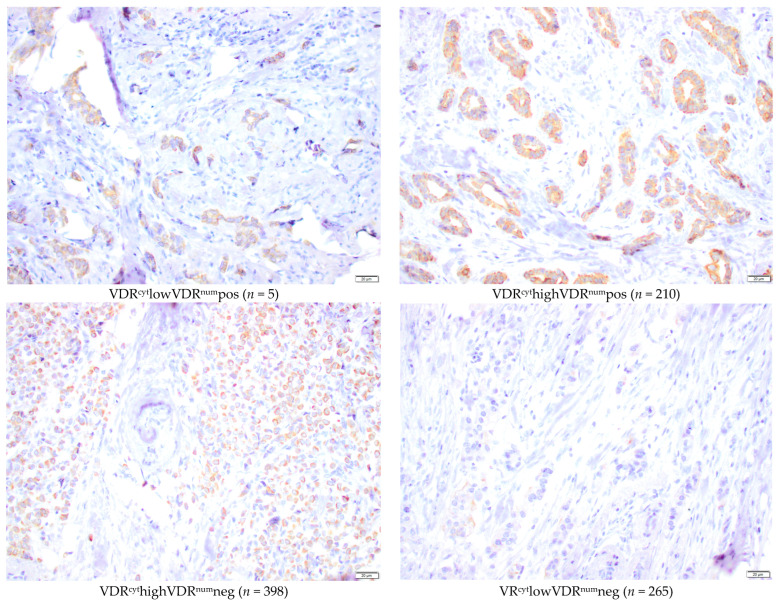
Microscopic representative images of immunohistochemical staining intensities of nuclear membrane and cytoplasmic VDR (40×) in the TMA. Bar represents 20 µm.

**Figure 3 nutrients-16-00931-f003:**
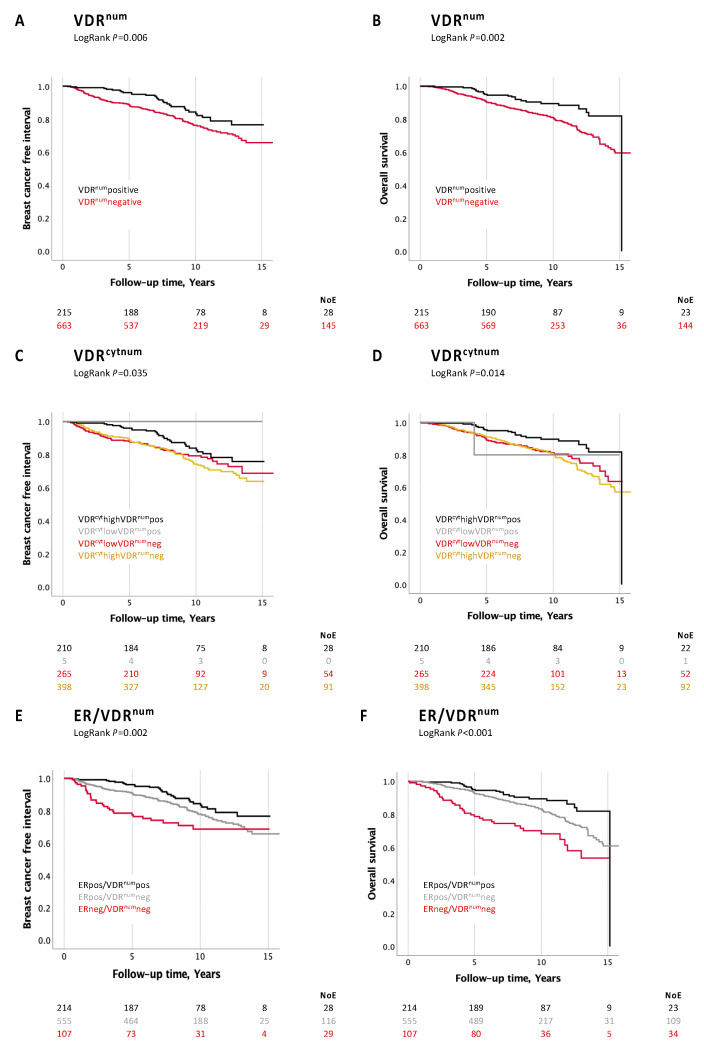
Kaplan–Meier estimates from univariable analyses. (**A**). VDR^num^ in relation to BCFI. (**B**). VDR^num^ in relation to OS. (**C**). VDR^cytnum^ in relation to BCFI. (**D**). VDR^cytnum^ in relation to OS. (**E**). ER/VDR^num^ in relation to BCFI. (**F**). ER/VDR^num^ in relation OS.

**Figure 4 nutrients-16-00931-f004:**
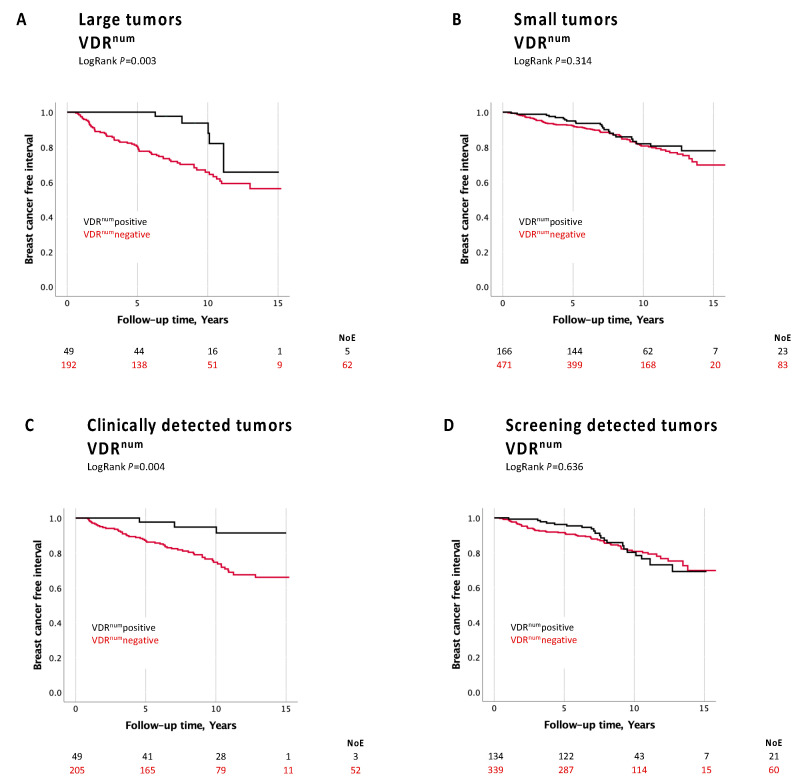
Kaplan–Meier estimates from univariable analysis of VDR^num^ stratified for tumor size and mode of detection in relation to BCFI. (**A**). Large tumors (pT2,3,4): VDR^num^ in relation to BCFI. (**B**). Small tumors (pT1): VDR^num^ in relation to BCFI. (**C**). Clinically detected tumors: VDR^num^ in relation to BCFI. (**D**). Screening detected tumors: VDR^num^ in relation to BCFI.

**Table 1 nutrients-16-00931-t001:** Distributions of tumor characteristics, treatment, and breast cancer events in relation to nuclear membrane VDR and number of patients with missing data for each variable.

Eligible Cases	All *n* = 1018				
Tumor in Tissue Microarray *n* (%)		Yes 984 (96.7)			No 34 (3.3)
VDR Assessable *n* (%)		Yes 878 (89.2)		No 106 (10.8)	
Nuclear Membrane VDR *n* (%)		Negative 663 (75.6)	Positive 215 (24.5)		
Factor	*n* (%)	*n* (%)	*n* (%)	*n* (%)	*n* (%)
Invasive tumor size					
1–20 mm	743 (73.0)	471 (71.0)	166 (77.2)	76 (71.7)	30 (88.2)
>20 mm *	275 (27.0)	192 (29.0)	49 (22.8)	30 (28.3)	4 (11.8)
Lymph node status					
Negative	627 (61.6)	390 (58.8)	135 (62.8)	79 (74.5)	23 (67.6)
Positive	389 (38.2)	271 (40.9)	80 (37.2)	27 (25.5)	11 (32.4)
Nottingham grade					
I	258 (25.3)	123 (18.6)	87 (40.5)	38 (35.8)	10 (29.4)
II	502 (49.3)	326 (49.2)	104 (48.4)	55 (51.9)	17 (50.0)
III	257 (25.2)	214 (32.3)	24 (11.2)	12 (11.3)	7 (20.6)
Histological type					
Ductal	824 (80.9)	554 (83.6)	176 (81.9)	67 (63.2)	27 (79.4)
Lobular	116 (11.4)	60 (9.0)	24 (11.2)	28 (26.4)	4 (11.8)
Other/mixed	78 (7.7)	49 (7.4)	15 (7.0)	11 (10.4)	3 (8.8)
ER status					
Pos (>10%)	894 (87.8)	555 (83.7)	214 (99.5)	97 (91.5)	28 (82.4)
PgR status					
Pos (>10%)	723 (71.0)	441 (66.5)	179 (83.3)	79 (74.5)	24 (70.6)
HER2					
Amplified	110 (11.5)	87 (13.5)	10 (4.7)	7 (9.0)	6 (31.6)
Unknown	63	18	2	28	15
Triple-negative					
Yes	74 (7.3)	68 (10.3)	0 (0)	5 (4.8)	1 (3.2)
Unknown	7	2	0	2	3
Treatment					
Final type of operation					
Mastectomy	410 (40.3)	271 (40.9)	85 (39.5)	40 (37.7)	14 (41.2)
Chemotherapy	258 (25.3)	193 (29.1)	41 (19.1)	15 (14.2)	9 (26.5)
Radiotherapy	644 (63.4)	418 (63.0)	139 (64.7)	68 (64.2)	19 (55.9)
Herceptin	73 (7.2)	54 (8.1)	10 (4.7)	4 (3.8)	5 (14.7)
Endocrine therapy **					
Tamoxifen	572 (64.0)	382 (68.2)	119 (55.6)	58 (59.8)	13 (46.4)
Aromatase inhibitors	371 (41.5)	249 (44.9)	80 (37.4)	32 (33.0)	10 (35.7)
Event					
Any breast cancer event	195 (19.2)	145 (21.9)	28 (13.0)	15 (14.2)	7 (20.6)
Death	188 (18.5)	144 (21.7)	23 (10.7)	15 (14.2)	6 (17.6)

Unknown presented when exceeding 0.5%. Percentages do not include missing categories. * Or muscular or skin involvement (pT2/3/4). ** Out of ER-positive *n* = 894.

**Table 2 nutrients-16-00931-t002:** Distribution of patient characteristics of all eligible cases in the cohort in relation to nuclear membrane VDR and number of patients with missing data for each variable.

Eligible Cases	All *n* = 1018				
Tumor in Tissue Microarray *n* (%)		Yes 984 (96.7)			No 34 (3.3)
VDR Assessable *n* (%)		Yes 878 (89.2)		No 106 (10.8)	
Nuclear Membrane VDR *n* (%)		Negative 663 (75.6)	Positive 215 (24.5)		
Factor	*n* (%)	*n* (%)	*n* (%)	*n* (%)	*n* (%)
Age at diagnosis					
≥50	816 (80.2)	535 (80.7)	168 (78.1)	89 (84.0)	24 (70.6)
BMI at inclusion					
≥25	503 (50.8)	336 (51.9)	100 (49.0)	49 (46.7)	18 (54.5)
Unknown	28	15	11	1	1
Waist circumference					
≥80 cm	731 (74.6)	490 (76.1)	143 (70.8)	73 (72.3)	25 (75.8)
Unknown	38	19	13	5	1
Total breast volume					
≥850 mL	492 (57.3)	335 (58.9)	87 (50.0)	54 (62.1)	16 (57.1)
Unknown	160	94	41	19	6
Parity					
Parous	896 (88.0)	576 (86.9)	196 (91.2)	93 (87.7)	31 (91.2)
Oral contraceptives					
Ever	722 (71.0)	467 (70.4)	151 (70.6)	77 (72.6)	27 (79.4)
Menopausal hormone therapy					
Ever	446 (43.9)	281 (42.4)	96 (45.1)	57 (53.8)	49 (35.3)
Alcohol abstainer					
Yes	106 (10.4)	72 (10.9)	23 (10.7)	11 (10.4)	0 (0)
Smoking currently					
Yes	206 (20.2)	130 (19.6)	41 (19.1)	31 (29.2)	4 (11.8)
Vitamin D supplements					
Yes	103 (10.2)	71 (10.8)	16 (7.5)	10 (9.4)	6 (17.6)
Unknown	7	6	1	0	0
Season of operation					
Winter (January–March)	251 (24.8)	160 (24.1)	58 (27.0)	28 (26.4)	5 (14.7)
Spring (April–June)	303 (29.8)	193 (29.1)	69 (32.1)	33 (31.1)	8 (23.5)
Summer (July–September)	177 (17.0)	111 (16.7)	38 (17.7)	20 (18.9)	8 (23.5)
Fall (October–December)	287 (28.4)	199 (30.0)	50 (23.3)	25 (23.5)	13 (38.2)
Detection mode (45–74 years)					
Clinical	290 (33.8)	208 (37.8)	52 (28.0)	21 (22.1)	9 (32.1)
Screening	569 (66.2)	342 (62.2)	134 (72.0)	74 (77.9)	19 (67.9)

Unknown presented when exceeding 0.5%. Percentages do not include missing categories.

**Table 3 nutrients-16-00931-t003:** Breast cancer free interval (BCFI) and overall survival (OS) in relation to nuclear membrane VDR status assessed using multivariable adjusted Cox regression analyses.

Breast Cancer Free Interval							
Nuclear Membrane VDR Level	Included in Analyses	Total *n*	Events *n*	HR ^1^ (CI 95%)	HR ^2^ (CI 95%)	HR ^3^ (CI 95%)	HR ^4^ (CI 95%)
Negative	Complete case *	663	145	ref	ref	ref	ref
Positive		215	28	0.61 (0.40–0.91)	0.71 (0.46–1.07)	0.72 (0.48–1.10)	0.70 (0.46–1.07)
Negative	All included **	730	155	ref	ref	ref	ref
Positive		288	40	0.64 (0.44–0.95)	0.74 (0.49–1.10)	0.75 (0.50–1.12)	0.71 (0.47–1.06)
**Overall Survival**							
Negative	Complete case *	663	144	ref	ref	ref	ref
Positive		215	23	0.51 (0.33–0.79)	0.63 (0.40–0.99)	0.69 (0.42–1.06)	0.67 (0.42–1.08)
Negative	All included **	764	161	ref	ref	ref	ref
Positive		254	27	0.52 (0.34–0.78)	0.62 (0.40–0.96)	0.64 (0.41–0.99)	0.65 (0.42–1.00)

* Only patients with no missing values included in analyses. ** Multiple imputation used for missing values; all patients in the cohort included in analyses. ^1^ Model 1: Adjusted for time between surgery and staining. ^2^ Model 2: Adjusted for time between surgery and staining, age, tumor size, node status, grade III. ^3^ Model 3: Adjusted for time between surgery and staining, age, tumor size, node status, grade III, ER status. ^4^ Model 4: Adjusted for time between surgery and staining, age, tumor size, node status, grade III, ER status, BMI, smoking, adjuvant treatments.

## Data Availability

The data that support the findings of this study are available on request from the corresponding authors (L.H. and H.J.). The data are not publicly available due to Swedish restrictions.

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
