# Peer review of "The Vitamin D Receptor as a Prognostic Marker in Breast Cancer—A Cohort Study"

_nutrients, 2024, doi:10.3390/nu16070931_

Round 1

Reviewer 1 Report

Comments and Suggestions for Authors

The manuscript by Huss et al describes the staining pattern of the vitamin D receptor (VDR) in a large cohort of women in Sweden, and shows associations with patient survival. It is largely well written and of interest. Although similar studies have been published (at least 7 studies), the advantage of this study is the size of the cohort (n=878 evaluated on TMAs) and the analysis of subcellular localisation of VDR.

However, the manuscript has several serious drawbacks that need to be addressed. The tables are very extensive and are missing headings for the columns, making it impossible to interpret. The entire premise of the study is based on immunohistochemical detection, and subcellular localisation, of VDR, yet no evidence (photos) are provided. The authors need to show examples of the different staining intensities (those shown in Fig 1 are insufficient, too small and poor resolution). The analysis of IHC staining appears to have been done manually, but details are sparse. Would TMAs be sufficient to capture the tumour heterogeneity of breast cancer tumours?

Detailed comments:

Abstract               The number of samples is misleading as only 878 of the 984 were evaluated.

M&M     Explain why 106 cores were non-evaluable for VDR

               How was the data of the individuals who had two separate tumours handled in the cohort?

               What is a waist circumference of 0.8 (unit?)

               How was the sensitivity and specificity of the Santa Cruz antibody assessed in this study? How was it optimised? What positive and negative controls were used?

               Why was overall survival (OS) and not disease-specific survival assessed?

               Tables are unclear; what is 100%? Column labels missing.

               It’s a shame that ethnicity data was not collected, but ethics and consents appear to be in order.

Results  Suggest dividing table 1A,B into tables 1 and 2; again missing column description

               Line 240: clarify what is meant here by ‘double negative’ tumours

               Are all ER- cores also VDR-?

               Why was data adjusted for ‘time between surgery and staining’? Is there evidence that cores have degraded?

               Table 2: suggest making OS heading bold with lines either side
Was this a multivariate analysis?

               Figure 2: Missing censored patients; this is usually shown with vertical short lines

Discussion           Is there a reason to have both tables in an appendix and supplementary?

References          All numbers are mixed up; please check automated referencing style before submitting.

Reviewer 2 Report

Comments and Suggestions for Authors

Authors found that positive VDR staining of the nuclear membrane in breast cancer is significantly associated with favorable tumor characteristics. And nuclear membrane levels of VDR is a better predictor of prognosis than cytoplasmic levels of VDR. The conclusion is interesting, and they propose a new diagnostic method for breast cancer. However, since the conclusion cannot be immediately understood just by looking at the figures.

Among the results in Table 1B, there are some combinations that are significantly associated with each other. But we can not know which ones are significant until we read the results. Is it possible to create a diagram that allows us to know significant association at a glance?

The staining images of VDR nuclear localization, nuclear membrane localization, and cytoplasmic localization are only shown in Figure1, and the differences cannot be clearly understood from Figure1. Typical staining patterns of VDRcytlowVDRnumpos, VDRcythighVDRnumpos, VDRcythighVDRnumneg, and VDRcytlowVDRnumneg should be clearly shown as enlarged images in the figures.  

Round 2

Reviewer 1 Report

Comments and Suggestions for Authors

The authors have significantly improved their manuscript.

Author Response

Thank you again for taking the time to review our work. We are very happy that you find improvement of the paper.

Kind Regards

Reviewer 2 Report

Comments and Suggestions for Authors

Authors need to write how authors performed the experiments in the figure legend. Basically, the legend is not written properly in either the table or the figure, so it is difficult to understand what the tables or figures show.

Author Response

Thank you for taking the time to once again review our work. We have now updated legends of figures and tables as followed:

Figure 1. Flowchart of inclusion and exclusion of patients and representative images of immunohistochemical staining intensities of nuclear membrane and cytoplasmic VDR (40x) in breast cancer tissue. The bar represents 20 mm.

 Table 1. Distribution of patient characteristics of all eligible cases in the cohort in relation to nuclear membrane VDR and number of patients with missing data for each variable.

 Figure 2.  Microscopic representative images of immunohistochemical staining intensities of nuclear membrane and cytoplasmic VDR (40x) in the TMA. Bar represents 20 µm.

Table 2. Distributions of tumor characteristics, treatment, and breast cancer events in relation to nuclear membrane VDR and number of patients with missing data for each variable.

 Table 3.  Breast cancer free interval (BCFI) and overall survival (OS) in relation to nuclear membrane VDR status assessed by multivariable adjusted Cox regression analyses.

 Figure 3. Kaplan-Meier estimates from univariable analyses of VDRnum, VDRcytnum and ER/ VDRnum in relation to BCFI and OS.

 Figure 4. Kaplan-Meier estimates from univariable analyses of VDRnum stratified by tumor size or mode of detection in relation to BCFI.

We hope that this will further improve our manuscript.

Kind Regards